Leveraging the potential of nature to meet net zero greenhouse gas emissions in Washington State

Robertson James C. jrobertson@tnc.org 1
Randrup Kristina V. 2
Howe Emily R. 1
Case Michael J. 1
Levin Phillip S. 1 3
1 The Nature Conservancy , Seattle , WA , United States of America
2 The University of Washington , Seattle , WA , United States of America
3 School of Marine and Environmental Affairs, The University of Washington , Seattle , WA , United States of America
Wang Xinfeng
Electronic publication date: 2021 Jul 21
Publication date: 2021
Volume: 9
Electronic Location ID: e11802
Received 2021 Mar 30; Accepted 2021 Jun 26
Copyright: ©2021 Robertson et al.
Copyright year: 2021
Copyright holder: Robertson et al.
License: This is an open access article distributed under the terms of the Creative Commons Attribution License, which permits unrestricted use, distribution, reproduction and adaptation in any medium and for any purpose provided that it is properly attributed. For attribution, the original author(s), title, publication source (PeerJ) and either DOI or URL of the article must be cited.
License URL: https://creativecommons.org/licenses/by/4.0/

Keywords: Climate change, Nature-based, Natural climate solutions, Mitigation, Justice, Equity, Distribution, Resiliency, Adaptation, Washington

Funding: No funding was received for this study.

==============================
The State of Washington, USA, has set a goal to reach net zero greenhouse gas emissions by 2050, the year around which the Intergovernmental Panel on Climate Change (IPCC) recommended we must limit global warming to 1.5 °C above that of pre-industrial times or face catastrophic changes. We employed existing approaches to calculate the potential for a suite of Natural Climate Solution (NCS) pathways to reduce Washington’s net emissions under three implementation scenarios: Limited, Moderate, and Ambitious. We found that NCS could reduce emissions between 4.3 and 8.8 MMT CO2eyr−1 in thirty-one years, accounting for 4% to 9% of the State’s net zero goal. These potential reductions largely rely on changing forest management practices on portions of private and public timber lands. We also mapped the distribution of each pathway’s Ambitious potential emissions reductions by county, revealing spatial clustering of high potential reductions in three regions closely tied to major business sectors: private industrial forestry in southwestern coastal forests, cropland agriculture in the Columbia Basin, and urban and rural development in the Puget Trough. Overall, potential emissions reductions are provided largely by a single pathway, Extended Timber Harvest Rotations, which mostly clusters in southwestern counties. However, mapping distribution of each of the other pathways reveals wider distribution of each pathway’s unique geographic relevance to support fair, just, and efficient deployment. Although the relative potential for a single pathway to contribute to statewide emissions reductions may be small, they could provide co-benefits to people, communities, economies, and nature for adaptation and resiliency across the state.

Introduction

The Intergovernmental Panel on Climate Change warned of significant impacts to economic, ecological and social systems if average global temperatures exceed 1.5 °C (IPCC, 2018). To prevent warming above this threshold, the world must reduce net emissions by 50% by 2030 and to zero by the middle of the century. Achieving this ambitious goal requires significant investment to reduce greenhouse gases emitted by the transportation, the built environment, and energy systems (Richter et al., 2008). Additionally, changes in land management, ecosystem restoration and conservation have the potential to increase carbon storage and avoid greenhouse gas (GHG) emissions across forests, wetlands, grasslands, and agricultural lands (Griscom et al., 2017). Such “Natural Climate Solutions” (NCS) can significantly contribute to CO2 equivalent (CO2e) emissions reduction goals at global (Griscom et al., 2017), national (Fargione et al., 2018), and regional levels (Cameron et al., 2017; Graves et al., 2020). Indeed, Griscom and colleagues (2017) suggested that if NCS were deployed in 2017, by 2030 they could mitigate over one-third of the GHG emissions needed to keep global warming below 2 °C.

The reduction of GHGs is a global problem, and associated goals are often promulgated at international or national scales (Gupta, 2010). Implementation of NCS is fundamentally a natural resource matter, and thus in the United States is managed by landowners and legal authorities acting within nested and overlapping federal, state, and county regulatory and jurisdictional frameworks. While national and international agreements are necessary to create accountability and to deploy broadscale decisions, top-down approaches can impede effective solutions when they do not consider equity across land-use sectors in cross-scale decision-making (Landauer, Juhola & Klein, 2019), consider indigenous and local community values (Scoones, 2016, Nightingale et al., 2020) or consider ecosystem-specific limitations (Bustamante et al., 2014). Thus, investigations of NCS must not only estimate the potential magnitude of emission reductions by various NCS approaches, but they must also reveal where on the landscape NCS opportunities exist. Understanding the geographic pattern of NCS opportunities is foundational to the equitable distribution of costs and benefits, the assessment of the social and political feasibility of different NCS strategies, and the assessment of social-ecological co-benefits of NCS implementation (Bustamante et al., 2014; Klinsky et al., 2017; Soto-Navarro et al., 2020).

In 2020, the Washington State (USA) legislature passed the Climate Pollutants Limits bill, which requires the state to reduce GHG emissions to 45% below 1990 levels by 2030, 70% below 1990 levels by 2040, and 95% below 1990 levels by 2050, plus offsets to address the final 5% to achieve carbon neutrality (Washington State Legislature, 2020). This is the first law in the U.S. that sets a net zero target and keeps Washington in line with the goals of the Paris Climate Accord agreement. Importantly, this law explicitly establishes a policy to promote the removal of carbon from the atmosphere through voluntary and incentive-based sequestration activities on natural and working lands. That is, the bill specifically calls for the development of NCS as part of the State’s net emissions reduction portfolio.

In this paper we aim to assess the degree to which NCS can contribute to Washington State’s decarbonization policy. Moreover, because the geographic distribution of NCS opportunities in Washington State has not been systematically evaluated, we identify where on the landscape NCS pathways could be implemented to maximize carbon sequestration and prospective co-benefits.

As a first step in connecting specific NCS pathways to their appropriate geographies, and thereby to ecological and socio-economic systems, we quantified potential NCS emissions reductions in Washington State at the scale of counties. We report the potential magnitude and geographic distribution of carbon reductions of a series of NCS pathways at multiple spatial scales and levels of intensity. Our goal is to provide the foundational spatial information needed to conduct inclusive planning and assessments for climate mitigation and adaptation.

Materials & Methods

Study context

The latest account of Washington statewide emissions was 99.57 MMT CO2e in 2018 (WECY, 2021). To achieve a net zero goal by 2050, reductions in emissions will have to occur directly within transportation, energy, agriculture, and industry, and the State also recognizes that increasing carbon sequestration rates and avoiding activities that continue GHG emissions in nature-based activities will be required (WECY, 2019; WA Governor’s Office, 2019).

In this paper, we draw on existing methods (Griscom et al., 2017; Cameron et al., 2017; Fargione et al., 2018; Graves et al., 2020) to calculate emissions reductions potential from Natural Climate Solutions in Washington and for each of the 39 counties within the state.

We chose to focus on the county scale because Washington counties are feasible units for analyzing industry clusters, or regionally concentrated groups of closely related industries that may fall in different business sectors but are closely linked by local commerce between them (Chrisinger, Fowler & Kleit, 2015). Regionalized climate solutions, including NCS, could be deployed within industry clusters to account for indirect and induced economic impacts as well as to create resilience for businesses involved. For example, the Washington maritime industry is implementing programs such as low emissions fleets and technical training to retain jobs in local communities (Washington State Department of Commerce, 2019). Moreover, county-level assessment is commonly used for social and resource-related objectives such as logistical deployment of state and federal programs like crop loss and disaster insurance (USDA NASS, 2020), annual timber harvest accounting (WDNR, 2018), state economic forecasts (Washington State Office of Financial Management, 2019), and equitable distribution of social programs and healthcare (Washington State Department of Health, 2018). County analysis units also allow finer-scaled socio-demographic analysis with US Census Tracts, Block Groups, and Blocks nested within their boundaries (U.S. Census Bureau, 2010).

General approach

We calculated the emissions reduction potential of 11 NCS pathways (here referred to as pathways) in Washington State, USA, largely following the approaches used previously in the states of Oregon (Graves et al., 2020) and California (Cameron et al., 2017), the United States nationally (Fargione et al., 2018), and the world (Griscom et al., 2017). The NCS pathways examined included activities associated with avoided land conversion, land management practices, and restoration (Table 1). In Washington State, the ecosystems of focus included temperate forestlands, shrub-steppe sagebrush lands, agricultural landscapes, riparian forests, and tidal wetlands.

Table 1 Descriptions of each NCS pathway adapted from Graves et al. (2020).

Pathway definitions were adapted from Cameron et al. (2017), Griscom et al. (2017), and Fargione et al. (2018).

Theme	Natural climate solution	Description	
Avoided Conversion	Avoided conversion of forests to rural development	Emissions avoided by limiting anthropogenic conversion of forests to low-density and agricultural development.	
Avoided conversion of forests to urban development	Emissions avoided by limiting anthropogenic conversion of forests to high-density, urban development.	
Avoided conversion of sagebrush-steppe to invasive annual grasses	Emissions avoided by limiting the conversion, post-fire, of sagebrush-steppe to invasive annual grasses; assumes active management of sagebrush-steppe recovery.	
Avoided conversion of grasslands to tilled cropland	Emissions avoided by limiting the anthropogenic conversion (e.g., tilling) of existing grassland to intensive agriculture.	
Land Management	Extended timber harvest rotations	Avoided emissions and increased sequestration associated with deferring harvest on a portion of Washington’s forest. We consider timber harvest across all forest ownerships in Washington but limit deferred harvest to counties with lower risk of wildfire.	
Use of cover crops	Increased carbon sequestration due to use of cover crops, either to replace fallow periods between main crops or as inter-row cover in specialty crops such as orchards, berries, and hops.	
No-till agriculture	Increased carbon sequestration due to the use of no-till agriculture on tilled cropland.	
Nutrient management	Avoided emissions from improving N fertilizer management on croplands, through reducing whole-field application or through variable rate application.	
Restoration	Replanting after wildfire on federal land	Increased carbon sequestration from increased post-wildfire reforestation on managed federal lands (e.g., wilderness areas are not included). This NCS assumes no salvage harvest or site-prep before replanting.	
Riparian forest restoration	Increased carbon sequestration through active replanting of forest along non-forested riparian areas.	
Tidal wetland restoration	Increased carbon sequestration due to restoring tidal processes where tidal wetlands were the historical natural ecosystem.	

Given the similarity of physiographic, industrial, and socio-economic characteristics of Washington and Oregon, we opted to assess 11 of the 12 pathways in Graves et al. (2020) excluding the sage brush restoration pathway because data are lacking for Washington, and rates of sage brush restoration are low. Griscom et al. (2017) and Fargione et al. (2018) investigated additional NCS pathways, but we opted not to include these because of their low relevance or low practicality in Washington. The Cascade Mountain Range divides both states into long Pacific Coastlines and wet temperate forests in the west and dry forests and shrub-steppe in the east of the state. Timber, agriculture, and tourism direct these states’ rural economies (Spies et al., 2019). However, the lands and waters making up the Puget Sound region, also called Puget Trough, contains a naturally protected inland sea unique to Washington, though many of its terrestrial ecosystems function similarly to coastal ecosystems and western forests of Oregon.

We conducted three effort-level scenarios of future NCS pathway implementation across a 31-year period of simulation, applied to a hypothetical timeframe of 2020 through 2050. However, we recognize that actual deployment will require some years of program development. We calculated reductions annually to quantify potential cumulative CO2e reductions over that period.

The three effort scenarios –Limited, Moderate, and Ambitious –highlight the potential for each pathway to reduce GHG emissions under different intensities of implementation relative to the variability of the pathway’s existing baseline rate of change, or historical variation. Scenarios differ by the degree to which implementation rates ramp up in the first decade and level off or continue to increase over the remainder of the 31-year study period (Table S1). Applying a ramp-up period represents a strategic approach to work hard at emissions reductions in the near-term so as to not have greater burden as we approach 2050. For each pathway within each scenario, a feasible implementation rate was determined via stakeholder feedback collected by Graves and colleagues (2020) in Oregon. Given the similarities of the Washington and Oregon timber industry history and socioeconomic changes following Northwest Forest Plan implementation in both states (Grinspoon, Jaworski & Phillips, 2016), feasibility estimates collected from Oregon stakeholders are reasonable for our study. Importantly, implementation rates of scenarios are meant to illustrate a range of potential outcomes for each NCS pathway.

All scenarios provide additionality, meaning they are equal to the prior year’s activity rate plus a calculated future implementation. We assume future implementation of each pathway will occur at some proportion of the historical variation of the pathway (e.g., annual variation of timber harvested from 2003 to 2017). Historical variation is calculated as the baseline annual rate of change times the historical Coefficient of Variation (CV). For pathways with input data provided in annual time series (Deferred Timber Harvest and Replanting After Wildfires), we calculated historical Coefficient of Variation (CV) over the available data years for each county and each region. When the CV was greater than the amount of activity, we capped the baseline rate at 100% of the activity to keep from calculating values greater than the available resource. For all other pathways, we assumed a conservative annual historical variation of 10% in place of the CV.

Scenarios for each pathway can be generalized as follows (see Table S1 for a more detailed description). The Limited scenario can be generalized as a steady increase by 10% of the historical variation each year, with growth reaching 100% of that historical variation after ten years of implementation and then remaining at 100% through the last twenty years (Tables S2 and S3). Moderate and Ambitious scenarios are generally characterized as the historical variation times a scenario growth coefficient for each pathway to reach a target rate during the first decade and then holding at or continuing to increase from that rate through the second and third decade (Tables S2 and S3). The initial growth coefficients ramp up implementation for the first ten years and are then replaced by lower growth coefficients or simply by the baseline rate, depending on the pathway, to simulate leveling off of the implementation rate during subsequent years.

We used Monte Carlo simulations to model the range of potential emissions reductions for all combinations of each county, NCS pathway (and sub-activity when necessary), simulation year, sequestration or avoided emissions rate, and implementation scenario (Tables S4 and S9). The simulations sample 1,000 iterations from a distribution created with the uncertainty range (i.e., standard deviation of sequestration or emission rate) for the pathway. All simulations and analyses were conducted using the software R (version 3.4.1) (Code S10 and S11).

NCS pathways

The following are generalized descriptions of the methods we used for each pathway. Pathways are organized under three themes: Land Management, Avoided Conversion, and Restoration. Detailed descriptions of the methods are provided in Graves et al. (2020), Tables S1–S3, and Code S10. We ran all three scenarios for each pathway.

Land management pathways

Extending Timber Harvest Rotations: We estimated business-as-usual annual carbon emissions with timber harvest data for each forest ownership in Washington from 2003 to 2017 (WDNR, 2018). Our assumed harvest deferral is from a 45-year rotation to a 75-year rotation to maximize sequestration potential from tree growth (Curtis, 1995). We performed calculations separately for areas east and west of the Cascade Mountain Range to address known productivity differences (Latta, Temesgen & Barrett, 2009) and excluded counties where wildfire risk was considered high or extreme on more than 50% of forestland (Gilbertson-Day et al., 2018) to limit influence of fire on carbon stocks. We also did not assess tribal lands in our analysis because their harvest volumes were not included in the published timber harvest reports.

To capture net carbon flux of this pathway, we calculated sequestration and emissions associated with harvest volume, below-ground biomass, unused mill residues, wood as commercial fuel, and short-lived (≤20-years) transformed wood products, as described in Graves et al. (2020). We developed these rates each for wet and dry dominated forests, west and east of the Cascades Mountain Range respectively. For delayed harvest scenarios, we also calculated added sequestration which occurs at higher rates on delayed harvest stands than on recently harvested stands of private-owned, even-aged managed forests (Smith et al., 2006). We used Monitoring Trends in Burn Severity (MTBS) fire perimeters (Finco et al., 2012) to filter out forest cover loss from wildfire and used growth tables for regional forests after clearcuts (Smith et al., 2006) to estimate differences in carbon sequestration in even-aged managed forests. We calculated baseline rates of clearcutting where private land boundaries intersect forest change data during the period 2000 to 2016 (Hansen et al., 2013, version 1.5).

Because Washington defines categories of private ownership differently than Oregon, we departed from Graves et al. in that we combined private industrial and private non-industrial forest ownerships into a single sub-activity. We therefore changed the scenario implementation rates to reflect this. Based on discussion with natural resource managers, we selected maximum possible implementation to be limited to 40% on private lands for the Ambitious scenario as a compromise between the percentages used by Graves et al. for private non-industrial forests (100%) and private industrial forests (21%). As we do not distinguish these two owner types, we chose to apply a value of 40% which largely biases towards the industrial. For state lands, we used the 32% used by Graves et al. Implementation is allowed to reach 100% on all other ownerships in western Washington counties and for all ownerships in eastern counties. After the first decade, the implementation holds at the maximum rates achieved in the first decade.

The Moderate scenario increases annually for the first decade by an amount equal to 10% of a percentage of the historical variation, reaching 100% of that allowed growth at ten years. These percentages are 30% of the baseline rate on private ownerships, 15% on state, and 75% on federal and other. During the last two decades, the rate holds steady at the maximum rate achieved over the first decade.

The Limited scenario differs in that it increases annually for the first decade by an amount equal to 10% of the historical variation, and then it drops and stabilizes at half the historical variation for the remaining twenty years. In some cases, the Limited scenario can result in emissions reductions greater than those of the Moderate scenario in a given year depending on the Moderate scenario’s proportional growth coefficient and the CV for the pathway’s baseline activity.

Agricultural Practices: To prevent deemphasis of the potential contribution of agriculture to NCS in Washington, we present Agricultural Practices as a single pathway though we calculated it as the sum of three separate sub-activities: applying cover crops, no-till practices, and nutrient management. The sub-activities were calculated independently of each other and do not simulate interactions between them that could potentially reduce or amplify GHG emissions, but we assume they will be applied with deliberate attention to address those interactions.

For cover crops, we estimated baseline rates of cover crop application with the 2012 and 2017 Census of Agriculture county-level data (USDA NASS, 2019a). Where county-level data were not made available, we calculated each of those county’s cover crop areas using the difference of the statewide reported cover crop area and the total proportion of that difference relative to the area reported for the other year. The area of maximum possible implementation is equal to the reported cropland area without cover crops, and crop type is not specified. We assume cover crops can be applied to all croplands, though we recognize this likely is not the case. The maximum possible implementation area was not reached in our scenarios.

Similarly, for no-till agricultural practices, we estimated baseline rates of no-till agriculture with data from the 2012 and 2017 Census of Agriculture county-level data (USDA NASS, 2019a). The area of maximum possible implementation is equal to the reported cropland area managed with tilling. Like cover crops, we assume no-till practices can be applied to all croplands although this is unlikely in practice. The maximum possible implementation area was not reached in our scenarios.

To calculate the nutrient management activity, we used published methods to produce county-level estimates of N fertilizer application, combining fertilizer sales and reported fertilizer chemical composition to convert tons of product sold to kg of N per county (Ruddy, Lorenz & Mueller, 2006; AAPFCO, 2014). This estimate disaggregates agricultural and other fertilizer uses. To maintain consistency with Graves et al. (2020), we use 40% of croplands as the maximum possible area of reduced fertilizer application.

Avoided conversion

Conversion of forests to development: We estimated the baseline rate of forest conversion on private land by intersecting land use data layers from 1994 and 2013, calculating conversion as the sum of two separate activities: from forests to urban (complete loss) and from forests to rural (partial loss) land uses (Gray et al., 2013; Gray et al., 2016). We grouped counties into east and west of the Cascade Mountain Range to address uncertainty in pre-conversion carbon stocks and sequestration.

Sagebrush-steppe conversion to invasive annual grasses: To estimate current rates of burn-associated conversion of sagebrush-steppe to invasive annual grasses, we combined the areal extent of fires from 1984 to 2014 (U.S. Geological Survey, 2018) with the annual grass dominance (U.S. Geological Survey, 2016). Background level of invasion by annual grasses is the proportion of burned areas dominated by invasive annual grasses minus the proportion of invasive annual grass-dominated land outside of burned areas.

Grassland conversion to cropland: We calculated rate of grassland loss in Washington from the 15,681 acres of land uncultivated since the 1970s which were retained or restored to grassland and subsequently converted to cropland between 2008 and 2012 (Lark, Salmon & Gibbs, 2015). Lacking county-level statistics, we used Lark, Salmon & Gibbs (2015) methods to classify grasslands with the 2012 USDA Cropland Data Layer (USDA NASS, 2019b), and then we calculated per county baseline conversion rates as proportional to the county’s percent of statewide grassland area.

Restoration

Riparian reforestation: We estimated the annual rates of riparian forest restoration using data on reported riparian restoration tree plantings from 1999 to 2015. The Washington Conservation Enhancement Reserve Program (CREP) 2015 monitoring summary report (Cochrane, 2016) served as our primary data source, and we supplemented this with acreage of riparian planting by county from the Washington State Recreation and Conservation Office PRISM (PRoject Information SysteM) database (WRCO, 2019). Because there is no mandatory system for reporting riparian reforestation efforts across Washington, we likely undercalculate the baseline riparian reforestation rate. Washington’s sequestration rate for riparian reforestation was derived as a weighted average of coastal and interior sequestration rates, where weights were calculated as the spatial distribution of efforts reported in the PRISM database. We assume a similar effort of implementation between PRISM and CREP in Washington.

Replanting after fires: We estimated the current level of replanting effort on US Forest Service and US Bureau of Land Management lands post-wildfire and quantified the C sequestration benefits from replanting versus natural regeneration on those lands (Smith et al., 2006). The baseline for replanting on Federal land was determined by examining current patterns of replanting after wildfire. In areas defined as high burn severity, the baseline rate is 0 hayr−1, as no replanting is occurring in these areas (U.S. Bureau of Land Management, 2018; U.S. Forest Service, 2019). Sequestration rates were calculated on a per year basis to simulate changes in growth rates over time.

Tidal wetland restoration: We used tidal wetland restoration areas and lost (converted) tidal wetland areas to estimate a baseline rate of restoration in most counties (PSMFC GIS, 2017; Ramirez, 2019). We assumed all lost tidal wetland area to have restoration potential, an area totaling 47,000 ha statewide, and each county’s potential restoration area was capped at its lost wetland area. We also assumed no further wetland loss will occur. In Puget Sound counties where tidal wetland restoration was not known or mapped in the restoration data, we assumed some restoration will occur in the future. We assigned each of those counties a restoration rate equal to its lost tidal wetlands hectares times a constant 0.00215, where that constant is the proportion of the statewide lost tidal wetland area that was restored annually in the Puget Sound during the baseline period. To estimate restoration rates in Pacific County, where known projects were not included in the input dataset, we obtained and inserted restoration areas described by the U.S. Fish & Wildlife Service (2015).

Results

Washington State

We estimate that NCS has potential to achieve an emissions rate of −8.8 MMT CO2eyr−1 in 31 years under an Ambitious pathway (Table 2). This is approximately 8.9% of theannual emissions reductions needed in Washington to achieve carbon neutrality by the year 2050 (assuming the baseline emissions rate from 2018 and start date of 2020). Of this 8.9% reduction, extending timber harvest rotations has the largest potential contribution (64%), followed by combined agricultural practices (16%) and avoided conversion of forests (13%).

Table 2 Assessed NCS pathways listed in descending order by their respective Ambitious potential CO2e reductions in the final year (ca. 2050) of a 30-year deployment for Washington state.

Reductions are shown here as the median of 1,000 Monte Carlo simulations along with the minimum and maximum ends of confidence intervals ranging from 5th to 95th percentiles.

 	Ambitious	Moderate	Limited	
Pathway	Median (MMT CO2e yr−1)	Median (MMT CO2e yr−1)	Median (MMT CO2e yr−1)	
	5th & 95th percentiles	5th & 95th percentiles	5th & 95th percentiles	
Extended timber harvest rotationa	−5.63	−3.46	−4.02	
−5.40	−5.84	−3.32	−3.60	−3.88	−4.16	
Agricultural practicesb	−1.40	−0.83	−0.10	
−1.28	−1.52	−0.75	−0.90	−0.0863	−0.1074	
Avoided conversion of forestc	−1.16	−0.58	−0.12	
−1.05	−1.27	−0.52	−0.64	−0.11	−0.13	
Riparian reforestation 	−0.29	−0.07	−0.01	
−0.29	−0.30	−0.07	−0.08	−0.01	−0.01	
Avoided conversion of sagebrush-steppe	−0.13	−0.03	−0.01	
−0.11	−0.14	−0.02	−0.02	−0.01	−0.01	
Tidal wetland restoration 	−0.11	−0.06	−0.03	
−0.10	−0.12	−0.05	−0.06	−0.03	−0.03	
Post-wildfire replanting (on federal land)	−0.11	−0.06	−0.03	
−0.09	−0.13	−0.05	−0.06	−0.03	−0.04	
Avoided conversion of grassland	<−0.01	<−0.01	<−0.01	
<−0.01	<−0.01	<−0.01	<−0.01	<−0.01	<−0.01	
Totald	−8.84	−5.10	−4.32	
−8.56	−9.10	−4.92	−5.25	−4.17	−4.46	
Notes.

a Combination of five activities: harvests on Private, State, Federal, and Other land owner types, and additional sequestration occurring where even-aged management would otherwise likely take place on Private timberlands.

b Combination of three activities: cover crops, no-till, and nutrient management on croplands.

c Combination of two activities: avoided conversion from forest-to-rural and forest-to-urban development.

d Median and confidence intervals of all pathways combined (called Total here) were calculated from all pathways simultaneously and not by summing the median and confidence interval results of each pathway listed in this table.

Our results demonstrate that Moderate and Limited scenarios offer pathways for reducing statewide emissions by approximately 5.1% and 4.3% respectively. Like the Ambitious scenario, extending timber harvest rotations, instituting different agricultural practices, and avoiding forest conversion make up the top pathways of these two scenarios. (Annual results of each combination of pathway, scenario, and geography –county and statewide –are in Tables S4–S8).

Geographic patterns

Potential NCS reductions are unevenly distributed across Washington (Fig. 1). Extending timber harvest rotations drives the spatial concentration of potential reductions toward coastal and southwest counties with large private timber ownership. Indeed, under the Ambitious scenario, that pathway’s −0.6 MMT CO2e potential 2050 reductions in Lewis County alone are greater than the highest aggregated reductions from all other pathways in any single county. Excluding the Extended Timber Harvest Rotations pathway, Grant County has the highest aggregated reductions, which are largely driven by cover crop and no-till activities in the Cropland Agriculture pathway and total −0.2 MMT CO2e potential reductions in 2050.

Figure 1 Prominence of the Extended Timber Harvest Rotations pathway among all NCS pathways in the state of Washington

(A) Total potential reductions from all pathways combined under the Ambitious scenario in 2050, shown with six equal classes in the displayed value range (minimum to maximum). (B) Total potential reductions under the Ambitious scenario in 2050 from all pathways except Extended Timber Harvest Rotations, shown with the same six classes as (A) though the actual range is smaller. Range minimum and maximum are the lowest and highest aggregated potential reductions of all Washington counties.

While the maximum potential CO2e reductions achievable via extending timber harvest rotations (and thus total NCS) is located in Washington’s western coastal counties, the maxima of other NCS pathways occurs in other regions of the state (Fig. 2). For instance, the maximum for the Avoided Conversion of Forests pathway occurs in the urbanizing Puget Sound region, and the greatest opportunities for the Cropland Agricultural Practices pathway is in agricultural regions in eastern Washington (Fig. 2).

Figure 2 Potential emissions reductions per county by each pathway in 2050, arranged by theme (Management, Avoided Conversion, Restoration).

Each map shows six equal classes in a given pathway’s stated range (minimum and maximum) of MMTCO2eyr−1 in 2050, and therefore ranges differ from map to map. County ID numbers match the county names in the included table. For example, Lewis County (ID 41) has the highest potential reductions of the Extended Timber Harvest Rotations pathway with a value of −0.6495 MMTCO2e, but Whitman County (ID 75) reductions are near zero. Conversely, reductions with Cropland Agriculture are near zero in Lewis County and highest in Whitman County with −0.1980 MMTCO2e, though that highest reduction is much less than Lewis County’s Extended Timber Harvest Rotations.

The greatest NCS gains can be clustered into three geographically concentrated resource-specific social-ecological systems: private industrial timber in southwestern wet forests, cropland agriculture in the Columbia Plateau and Palouse Prairie, and suburban development and urban sprawl in central Puget Sound (Fig. 3). Private industrial timber drives the Extended Timber Harvest Rotations pathway, with over half the emissions reductions potential in southwestern wet forests (plus the highly industrial forests of Stevens County in the northeast.) The second highest NCS reductions are found through Cropland Agricultural Practices pathway, with nearly half of its reduction potential in four eastern Washington counties. The Avoided Conversion of Forests pathway provides the third highest potential reductions with nearly half being clustered in five of Puget Sound’s fastest growing counties marked by suburban and rural development.

Figure 3 Map revealing spatial clusters where the highest-potential counties of the three highest-potential pathways provide approximately half of each of those pathway’s emissions reductions in 2050.

These NCS clusters are largely driven by major industry sectors within the region.

Discussion

Our estimates of the potential CO2e reductions with NCS ranged from 4.3 to 8.8 MMT CO2e by the 31st year of implementation, depending on the aggressiveness of our scenarios. These values are similar to the 2.9 to 9.8 MMT CO2e at the 31st year estimated by Graves et al. (2020) for Oregon and highlight the potential for NCS to contribute to atmospheric carbon sequestration efforts. Our results reveal that an ambitious implementation of NCS has the potential to achieve up to 8.9% of Washington’s net zero goal by 2050, with much of these gains achieved by extending timber harvest rotations. However, the three major pathways –Extended Timber Harvest Rotations, Cropland Agriculture, and Avoided Forest Conversion, have vastly different reduction potentials depending on where they are implemented across the state.

Similar to studies in other western US states, such as Oregon and California (Cameron et al., 2017; Graves et al., 2020), we found that extending harvest rotations on industrial forestlands in Washington’s wet forests offered the largest NCS contribution. Although current industrial forest management is focused on short-term capital gains (Lacy, 2006), we demonstrate that extending timber harvest rotations could have significant reductions in Washington’s GHG emissions, a finding that aligns with evidence showing that large trees disproportionately drive forest carbon cycle dynamics (Mildrexler et al., 2020). These reductions would likely supplement a list of co-benefits including improved water quality, improved summer baseflows, wildlife habitat, and salmon productivity.

Cropland agricultural pathways, such as cover crops, no-till agriculture, and nutrient management, offer the greatest climate mitigation potential for NCS in the eastern half of Washington where highly productive and extensive croplands are found. Historically, nutrient-rich shrub steppe and grassland ecosystems dominated these current agricultural landscapes due to frequent wildfires and a dry continental climate which limit forest productivity.

Our study also shines a spotlight on the substantial threat of land-use change and forest conversion in the rapidly growing Puget Trough, driven by a growing technology industry and the desire for affordable housing. However, recent analysis by the Washington Department of Fish & Wildlife and the Puget Sound Partnership found that conversion of forest cover loss to development has declined considerably and continuously in the region since 1991 (Pierce Jr et al., 2017; Puget Sound Partnership, 2020). The cause of this decline is not well understood, but slowdowns of development due to economic downturns and other possibly temporary factors may indicate that development and therefore conversion rates could increase with further economic growth. If the existing declining trend in forest conversion rates continues, this study may over-calculate the NCS potential of avoiding forest conversion in Washington State, though the net emissions reductions would likely still occur with or without implementing NCS under our scenarios.

Spatial patterns

Three factors drive the spatial patterns we report; (1) spatial extent of ecosystems; (2) abundance and productivity of vegetation; and, (3) land ownership. Extensive forest ecosystems across Washington State mean that NCS pathways that involve forest management could contribute substantially to NCS potential simply due to scale. It is thus not surprising that, combined, forest strategies (extending timber harvest rotations, forest avoided conversion, riparian reforestation, and post-wildfire replanting) contribute over 80% of the NCS potential in Washington State. Our Tidal Wetland Restoration results further demonstrate that natural ecosystem size strongly influences NCS sequestration potential. The Tidal Wetland Restoration pathway has its greatest potential for emissions reductions where restorable wetland area is currently large in spatial scale, such as the large estuarine deltas of eastern Puget Trough and southwest coastal Washington counties. By contrast, topographically constrained estuaries in the western Puget Trough and along the Olympic coast are smaller in size and therefore have lower NCS potential.

Geologic and climatic patterns drive the composition, abundance, and productivity of vegetation, thereby influencing the potential of NCS across Washington. While some regions, such as the moist forests along the coast are characterized by highly productive forests and high rates of carbon sequestration, dry areas of the state have less productive forests and are at high risk of wildfire. By excluding counties with high wildfire risk from our extended timber harvest rotations analysis, the geographic pattern is focused on counties with high productivity and low fire risk to west of the Cascade Mountain Range. Cropland agricultural pathways show distinct patterns as well, focusing where historically available water and fertile soils promote productive cropland in valleys of eastern Washington and floodplains of the Puget Trough.

Land ownership is a third major driver of geographic patterns. In Washington, public lands have low risk of forest conversion (Gray et al., 2013), and major changes to public forest management are limited by policy restrictions on federally-owned forests (Spies et al., 2019) and legally bound fiduciary obligations in the case of State Forest Lands trust (Washington Department of Natural Resources, 2021). Thus, potential for additional emissions reductions is generally highest in counties with large private ownership of forests such as Lewis and Stevens Counties. For those same reasons, counties with little private forest ownership likely will have lower relative NCS potential even if they have vast public forest ownership, such as Jefferson and Whatcom Counties with large state forests, US Forest Service ownership, and the Olympic and North Cascades National Parks respectively. This is particularly evident in Puget Trough counties, where remaining forests are generally public-owned. However, the remaining privately owned Puget Trough forests are more profitable for suburban development than for logging. As a result, the Avoided Conversion of Forests pathway holds its greatest potential in those Puget Trough counties.

Success and challenges

Success of a net zero goal likely requires individuals and decision-makers to redefine their notions of practicality, considering many pathways as courageous enterprises rather than unfeasible concepts. Even mitigation pathways that sequester relatively small amounts of carbon should be considered for at least two reasons. First, every small reduction will be necessary to approach net zero emissions, and negative emissions could help offset other states that do not achieve net zero emissions. Second, all pathways have co-benefits, and these co-benefits may contribute to climate adaptation and resilience in a region.

The scale at which decisions are made can inhibit the effective implementation and outcomes of climate mitigation strategies. Extensive actions addressing climate mitigation may be impeded by social and cultural differences as well as differences in lived experiences that are not considered in the broad-scale top-down decision-making approaches (Landauer, Juhola & Klein, 2019; Nightingale et al., 2020). Additionally, conflicting policy-objectives among different scales of governance can hinder adoption of broad-scale climate mitigation measures. Furthermore, the large scales at which mitigation is often discussed can be disconnected from the local scale of adaptation. Such separation may thwart efforts to build local support for mitigation and distances government accountability for addressing local needs (Nightingale et al., 2020).

The need for multiple solutions to the climate crisis is clear, and thus NCS should not be considered as an alternative to other carbon mitigation strategies (Anderson et al., 2019). While forest restoration and afforestation may be highly effective strategies for mitigating climate change (Bastin et al., 2019), these strategies alone cannot adequately nullify emissions from fossil-fuels and other anthropogenic GHGs at a global scale (Anderson et al., 2019; Friedlingstein et al., 2019). Indeed, our results highlight the potential for improved management of ecosystems and natural resources to provide greatest net emissions reductions among NCS pathways in Washington State. Also, while the costs of NCS can be low relative to other strategies (Griscom et al., 2017), a number of constraints can diminish practicality and effectiveness of natural solutions, such as political interests, local economic factors, varying scales of decision-making, deployment time, and undesired collateral effects such as resource leakage and biophysical changes in the environment (Nesshöver et al., 2017; Landauer, Juhola & Klein, 2019; Friedlingstein et al., 2019; Mulligan et al., 2020).

Logistics of NCS deployment and concerns about economic losses by industries and communities may also stall implementation of NCS. For example, an argument might be made that the economic cost of extending harvest rotations is too great, rendering this NCS pathway infeasible. Moreover, extending timber rotations can disrupt markets and undermine NCS benefits if the equivalent carbon sequestration is lost through harvesting elsewhere (i.e., leakage). Although leakage may limit this pathway’s practicality (Mulligan et al., 2020), in the Pacific Northwest leakage has been shown to be relatively small when compared to the substantial amount of carbon stored through improved forest management and embodied carbon in wood products (Diaz et al., 2018). While there is no doubt that implementation of this pathway may be challenging, even under our Ambitious scenario, only up to 40% of private timber harvest is subjected to lengthened harvest rotations, and many counties do not reach that maximum in our analysis. Thus, the utility of lengthening timber harvest rotations may be greater in Washington than in other locations.

The effects of leakage with NCS is a key consideration. All pathways have potential for leakage if not implemented appropriately, and NCS in theory and in practice should address this challenge. Just as NCS forest management needs to account for possible losses of carbon sequestered by other forests that may be logged to meet the demand for timber, agricultural practices and wetland restoration need to account for possible reductions in crop production that may increase elsewhere to meet market demand, thus diminishing mitigation gains. The inherent nature of agricultural management pathways for carbon is that they should retain or increase crop productivity through improvements to soil health, water efficiency, and other advantages (Dabney et al., 2010). While there are many challenges in implementing NCS, these obstacles may be relieved by cost efficiencies and other advantages arising from the focused spatial patterns, concentrated business sectors, and industry clusters associated with NCS pathways. Efficiencies and co-benefits might also be maximized when addressing multiple overlapping pathways in a region with a whole systems approach. Certainly, this is the case for sub-activities of the Cropland Agriculture pathway, where carbon benefits can be best realized through simultaneous management interventions that increase no-till cover crop practices and decrease N-fertilizer application (Stöckle et al., 2012; Beach et al., 2018; Schmidt et al., 2018).

With limited resources and regionalized economies, our results highlight that a fruitful way forward may be to deploy NCS at a county scale. With county-level information, state decision-makers can focus attention on regional issues and include local stakeholders in a manner that may be impractical statewide. As a result, appropriate program incentives addressing socio-cultural sensitivity, economic development, and ecological protections can be catered to the needs of specific communities and industries. In turn, those local stakeholders can work with state decision-makers to develop ecological and human resilience plans in concert with mitigation activities. The success of the local climate planning approach to climate policy implementation and achievement of implementation goals has yet to be widely studied, but early investigations from the European Union suggest climate change issues are best tackled by either (1) developing dedicated local plans in parallel with larger scale mainstream plans, or (2) starting with a dedicated local plan and subsequently developing a mainstream plan (Reckien et al., 2019).

Further research should look at the value per unit of pathway implementation to plan for the most cost-effective and optimal outcomes of implementing multiple pathways within a county. A simple calculation of mitigation potential per county area may provide a starting point (Table S9), but calculating the implementation per available resource quantity within the county could provide better means for prioritizing pathway deployment.

Multiple benefits of NCS

Despite real and potential benefits of NCS for climate change mitigation, NCS could be of greater service for climate resilience and other co-benefits (Griscom et al., 2017; Seddon et al., 2019) when implemented to avoid negative or unintended consequences (Hashida et al., 2020).

For example, numerous human health, social equity, and ecological benefits exist from reduced N fertilizer use. Nitrates from fertilizer seep into groundwater, contaminating human drinking water reservoirs (Keeler & Polasky, 2014). This nitrate contaminated water has been linked directly to high cancer rates, low fertility rates, increased water treatment costs, and devaluation of property in rural communities, including in Washington (Townsend et al., 2003; Dubrovsky et al., 2010; Moore et al., 2011; Keeler & Polasky, 2014). Likewise, reduction of N fertilizer use has benefits for marine and freshwater biota and soil health (Culman et al., 2010; Dubrovsky et al., 2010). NCS can generate multiple benefits when implemented in concert across whole systems. We found that much of the greatest emissions reduction potential by three pathways (Avoided Conversion of Forest, Riparian Restoration, and Tidal Wetland Restoration) exist in the Puget Trough, particularly overlapping in Snohomish County. The links between the upland forests, river corridors, floodplains, and estuaries are important ecologically, culturally, socially, and economically. From an ecological perspective, Chinook salmon (Oncorhynchus tshawytscha) are listed under the US Endangered Species Act and vital to the health of many freshwater and riparian ecosystems in the Pacific Northwest (Willson & Halupka, 1995). Chinook salmon use the full Snohomish River system, spawning in upland streams, rearing in mainstem floodplain and estuarine channels, feeding in nearshore marine waters, and migrating back upstream to spawn. These life stage requirements thus link Chinook salmon to the Avoided Conversion of Forest, Riparian Restoration, and Tidal Wetland Restoration NCS pathways. Upland forests help control stream flashiness and sedimentation, which can inhibit spawning and rearing (Beechie et al., 2013). Riparian tree cover cools streams, provides refugia along riverbanks, and delivers terrestrial insect food resources (Seavy et al., 2009; Beechie et al., 2013). High productivity tidal wetlands provide juvenile Chinook with the low salinity habitat they need to transition from fresh to saltwater living, as well as the saltmarshes which provide energy-rich food resources critical to early marine survival for juvenile Chinook (Simenstad & Cordell, 2000; David et al., 2014). Chinook salmon are also an important cultural and nutritional resource to the indigenous Coast Salish peoples, and fishing of the Chinook is an intrinsic right legally affirmed by the US Supreme Court’s 1974 Boldt Decision (United States v. Washington, 1974). In addition to improving conditions for Chinook salmon, these three pathways may provide other co-benefits such as mitigation for flooding of croplands and homes downstream, and improved water quality associated with sedimentation, nutrification, and fecal pathogens (DeGasperi et al., 2018).

Conclusions

Natural Climate Solutions have the potential to play a significant part in climate change mitigation. Our findings show that NCS can assist in achieving a Net Zero goal by 2050 in Washington, and those achievements range by the aggressiveness of NCS pathway implementation. Further research assessing costs and practicality of NCS and the ongoing impacts of climate change will yield additional insight that could influence the feasibility of NCS pathways. Likewise, rigorous examination of the ecological, social, cultural and health co-benefits of NCS is crucial for assessing the benefit-cost ratio of any pathway.

Clearly, a broad portfolio of tools and approaches will be needed to combat climate change. Indeed, to achieve carbon neutrality in Washington by mid-century, NCS pathways may be required even if reductions through changes in transportation, industry, and residential sectors can be achieved. Our work provides needed information that emphasizes the potential scope of NCS to achieve carbon goals as well as the spatial distribution of NCS opportunities. This is the foundation required to generate natural climate solutions that are fair, just and contribute significantly to address climate change in Washington and beyond. Addressing climate change is urgent.

Supplemental Information

Supplemental Information 1 Description of scenario details for calculating each pathway

A data dictionary is included as a tab in this file.

Click here for additional data file.

Supplemental Information 2 Values used as inputs to the R code, including baseline rates of each pathway activity, coefficients of variation for Monte Carlo simulations, and sequestration rates of each activity per specified baseline unit. Actual input files are included with the c

A data dictionary is included as a tab in this file.

Click here for additional data file.

Supplemental Information 3 Median and confidence interval outputs calculated statewide for each combination of scenario and year

A data dictionary is included as a tab in this file.

Click here for additional data file.

Supplemental Information 4 Evaluated results of Monte Carlo runs on each scenario in each year, statewide

A data dictionary is included as a tab in this file.

Click here for additional data file.

Supplemental Information 5 Evaluated results of Monte Carlo runs on each pathway activity per scenario in each year, statewide

Click here for additional data file.

Supplemental Information 6 Evaluated results of Monte Carlo runs on each scenario in each year, per county

A data dictionary is included as a tab in this file.

Click here for additional data file.

Supplemental Information 7 Evaluated results of Monte Carlo runs on each pathway subactivity per scenario in each year, per county

A data dictionary is included as a tab in this file.

Click here for additional data file.

Supplemental Information 8 Evaluated results of Monte Carlo runs on each pathway activity per scenario in each year, per county

A data dictionary is included as a tab in this file.

Click here for additional data file.

Supplemental Information 9 Emissions reductions per area of each county in final year of model, calculated from results in S8

A data dictionary is included as a tab in this file.

Click here for additional data file.

Supplemental Information 10 R code for estimating potential greenhouse emissions reductions in Washington using Natural Climate Solutions

The main code which outputs each Monte Carlo iteration and a compilation of those iterations to produce mean and low and high confidence intervals depending on the desired aggregation of NCS pathway, scenario, year, and geography.

Click here for additional data file.

Thanks to Rose Graves, Ryan Haugo, Michael Schindel, and Bryce Kellogg for sharing methods, data, and advice, to Mary Ramirez and Jennifer Burke for sharing data and thoughts on estuary restoration rates, and to Juliana Tadano, Elizabeth Matteri, Chase Puentes, and Pascale Chamberland for literature research. Thanks also to Ailene Ettinger and Kristina Bartowitz for detailed reviews, to Maia Murphy–Williams for facilitating some aspects of this project, and to Mo McBroom for early encouragement and insight.

Additional Information and Declarations

Competing Interests

Author Contributions

Data Availability

The authors declare there are no competing interests.

James C. Robertson conceived and designed the experiments, performed the experiments, analyzed the data, prepared figures and/or tables, authored or reviewed drafts of the paper, and approved the final draft.

Kristina V. Randrup performed the experiments, analyzed the data, authored or reviewed drafts of the paper, and approved the final draft.

Emily R. Howe, Michael J. Case and Phillip S. Levin analyzed the data, authored or reviewed drafts of the paper, and approved the final draft.

The following information was supplied regarding data availability:

Raw data and code are available in the Supplemental Files.

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
