# Peer review of "Leveraging the potential of nature to meet net zero greenhouse gas emissions in Washington State"

_PeerJ, doi:10.7717/peerj.11802_

## Round 0.1 · original submission · Minor Revisions

Based on the comments from the anomymous reviewers, your article requires minor revisions before it is accepted. Please try to submit the revised version within three weeks.

·

Basic reporting

Clear, unambiguous, professional English language used throughout.

Yes, the writing is nice and clear.

Intro & background to show context.

Yes, the background is clearly and efficiently explained.

Literature well referenced & relevant.

Yes, in general the appropriate literature is cited, see a below a few specific suggestions for improvements in the Methods section.

Structure conforms to PeerJ standards, discipline norm, or improved for clarity.

The structure is clear and appropriate.

Figures are relevant, high quality, well labelled & described.

See comments immediately below and elsewhere for improvements that should be made to figures and tables.

Table 2. Present only one place to the right of the decimal point.
Table 2. Footnote B is redundant.
Table 2. Use superscripts for “yr-1” throughout.

Figures 1 and 2. Present only one place to the right of the decimal point.
Figures 1 and 2. Scale results to area rather than totals per county (because the counties have different areas).
Raw data supplied (see PeerJ policy).
I cannot evaluate the extent to which data are provided, but there is R code and some supplemental tables.

Experimental design

Original primary research within Scope of the journal.

I have limited knowledge of the scope of this journal, but the manuscript contains valuable analysis and results that are worthy of publication in an applied environmental science journal of good quality.

Research question well defined, relevant & meaningful. It is stated how the research fills an identified knowledge gap.

Yes.

Rigorous investigation performed to a high technical & ethical standard.

Yes, after making the changes recommended in this review.

Methods described with sufficient detail & information to replicate.

See comments below for suggested modest improvements.

Line 92. Provide the GWP values used to generate the total of 99.87 MMTCO2e and provide the breakdown among gases, at least CO2, N2O, CH4 and other.

Lines 147 to 163. Annual mitigation for future scenarios seems to be calculated as a fraction of existing activity. How are biophysical or other limits to total mitigation included? For example, no-till can only be performed on some fraction of tilled acres, or possibly the total of tilled acres, but no more than that. For example, lines 199 to 207 seem to discuss limits on total area for deferred forest harvest, but it is not really clear how the annual rates are limited not to exceed the total area.

Lines 208 to 220, More information must be provided to clarify how the analysis was done. What area of what type of cropland was assumed to be suitable for cover crops? How is this limit applied in relation to estimates of annual mitigation? Are cover crops used where irrigation s used? Are they only winter cover crops, and if so, what about winter crops? Similarly for no-till, what area is deemed to be suitable for which crops? How is this limit used in relation to annual mitigation rates?

Lines 2017 to 2020. Fertilizer sales data from AAPFCO and related sources are often quite variable at the state level from year to year, and probably even more variable at the county level. For national scale analysis, such variation is not a problem or only a small one, but it is a bigger issue at State and even bigger at county scales. Was this variation deemed to be real, meaning that it reflects actual fertilizer application to soils in that year? How does this relate to estimates of N rates for major crops from other sources such as USDA-ERS? IF they are not already using it (if so, it must be cited) I suggest the authors investigate the use of the following data source, which provides estimates at the county scale. But this source doesn’t really solve the issues I raise, because it relies on the same AAFPCO data. Also, explain whether fertilizer use was only on farms or total including nonfarm use. Note that the source below may provide a breakdown of farm and nonfarm.

Brakebill, J. W., and J. M. Gronberg. 2017. County-Level Estimates of Nitrogen and Phosphorus from Commercial Fertilizer for the Conterminous United States, 1987-2012. U.S. Geological Survey.

Lines 219 and 220. How was nutrient use efficiency calculated? Do you mean recommend N rates for specific crops? Please clarify and explain thoroughly.

Lines 253-259. Explain how sequestration rates were estimated for both replanting and natural regeneration. For example, if from Smith et al. 2006. repeat that citation here.

Line 270. Explain and cite what is meant by “existing literature”.

It would be helpful to have some summary of the Ambitious, Moderate and Limited scenarios, perhaps in the form of a figure showing the time course of mitigation for each over the 30 year period.

Validity of the findings

The findings are valid. See below for improvements that should be made, or are recommended to be made to improve the results, discussion, and conclusions.

Line 274. Here and throughout the manuscript, report MMTCO2e values to only a single place to the right of the decimal, or perhaps 2 places when needed to maintain 2 significant digits.

Presenting results by county is OK, but the figures should present results weighted by county area, and the results should present results weighted by county area as well, referring to the revised figures. The existing county-based figures (unweighted by area) could be moved to the supplemental materials.

In the results section, It would be very valuable to also add tables and to discuss results weighted by the area of the actual mitigation activity, which of course is always less than the county area, sometimes quite a lot less. This would help clarify which practices are most effective per area versus those that are effective because they cover a lot of area. This could really help guide implementation of the practices. Probably the spatial patterns you report are simply a result of the area for which a practice can be implemented, rather than variation in the benefit for an area that has the mitigation practice. If this is true, it would be very useful information for implementation, if it is not true, then disaggregating these two actors for each practice would be very helpful. This analysis would support improved and more valuable discussion in the Spatial Patterns section beginning on line 344.

Lines 318-320. See above suggestion about efficacy of mitigation per hectare vs number of hectares. Clarify this statement to distinguish whether the mitigation efficacy for a practice varies in different locations in the state or whether the variation is due to the number of hectares for the practice in different regions.

Lines 411-412. Discuss more thoroughly the implications of extending the forest harvest rotation. Specifically, discuss whether this possible leakage would likely negate much or all of the GHG benefit, this is very important because it is the largest total GHG mitigation opportunity found in your analysis. If it is likely to be negated largely or entirely by leakage, that is very important to know. Climate change is a global issue, so not at least discussing the likely implications greatly weakens the manuscript. Leakage should also be discussed for each mitigation pathway, even if just to say that it is unlikely to be an issue for some reasons you can provide. For example, the agricultural practices if properly implemented in appropriate climatic, soil, and cropping conditions, should not reduce crop yields, and therefore should not cause leakage.

Lines 386-387. I suggest mentioning other co-benefits in addition to climate benefits, such as for air quality, water quality, and biodiversity. These have been included in other publications about Natural Climate Solutions, and are important because these benefits can be valuable and can help motivate mitigation, because the mitigation provides these other valuable benefits. This topic is important to balance the existing discussion of challenges to implementation. I also suggest adding a paragraph of discussion of these other important co-benefits to the section beginning on Line 438, The current discussion of Chinook salmon is very good, but other important co-benefits should be mentioned, especially in the context of providing evidence of social benefits that can exceed the cost of implementation.

Line 466. It would be very helpful to add a paragraph of discussion about costs versus benefits, even if it is only partly quantitative. Previous NCS publications have provided some cost estimates for different practices. Any information about the value of the benefits, including but not limited to the social cost of carbon, would help raise the important issue of whether it is beneficial for society to implement these practices, given the likely costs and benefits. This new paragraph would provide support for the statement on lines 43-474, or even support a stronger statement that there is evidence in many cases that the benefits could exceed the costs for at least some mitigation opportunities.

All underlying data have been provided; they are robust, statistically sound, & controlled.

I can’t fully evaluate this question, although the authors do provide substantive information in supplemental tables and provide R code. The authors should assure that all variables in supplemental tables are defined. For example, in supplemental Table 3 and others, what does the unlabeled Column A mean?

Also, please be sure to cite supplemental tables S3 to S7 in the manuscript, and where are their contents explained?

Speculation is welcome, but should be identified as such.

The results and discussion are appropriate.

Conclusions are well stated, linked to original research question & limited to supporting results

The conclusions are suitable, see minor suggested improvement above.

Additional comments

This manuscript is a valuable contribution to the literature after making the changes recommend above. The content and length are appropriate and it is well written.

Reviewer 2 ·

Basic reporting

No concerns. A few very minor suggestions to clarify the text:

Abstract: I don’t believe IPCC set 2050 as an explicit target for net zero greenhouse gas emissions. The actual signed UN Paris Agreement gives a vaguer “mid-century” timeline.

Ln 40: not “if global temperatures rise above 1.5C” but rather “if average global warming exceeds 1.5C”

Ln 42: not “reduce emissions” but “reduce net emissions”

Ln 76: “the State’s net emission reduction portfolio” – especially since this sentence is about removals

Ln 102-105: The sentence starting with “Regionalized climate solutions, including NCS, could be deployed within industry clusters…” is vague. Can the authors provide more detail or perhaps a detail of what this looks like beyond just citing the maritime industry?

Line 114: Griscom et al. (global) and Fargione et al. (US) include ~ 20 different NCS. The authors state that they used 11 of the 12 pathways in Graves et al. 2020 and justify the exclusion of sage brush. Can the authors further clarify why they did not include the other 8-10 NCS for those not familiar with the Washington landscape. Although this paper is clearly intended for a WA audience, there are methods and results in here that could be extended to other geographies.

Line 127: What does “their” refer to in “tourism direct their rural economies”?

Experimental design

There are many methodological details that need clarification. I flag several below, but I encourage the authors to go through the methods again and provide more references, justification, and details to allow readers to understand the methods.

• It’s possible with enough time I could follow the details in the R script and sort out the actual methods so I appreciate the clear time and effort that went into documenting input data, providing scripts, etc in the supplementary material. However, if the authors include scripts etc, then some clean-up is merited. For example, the provided script references files that I could not locate (e.g., model_carbon_reforestation_low.csv). There is an “activity” value in Table S1 that could potentially be used to subset and create csv inputs, but that is error prone. I applaud the authors efforts to be fully transparent, but they have not reached that point.

• Line 132-133: The authors state that “actual deployment will require some years of program development.” Can they clarify how implementation is temporally modeled? Given that we are now in 2021, a start year later than 2020 seems warranted. Methods are not currently clear how ramp up is incorporated (line 159 and beyond).

• Line 155: The “Limited” scenario should probably be called a BAU scenario. “Limited” implies some additional implementation has occurred (and NCS must be “additional” per Griscom et al. 2017 etc) but the limited scenario simply incorporates variation into a baseline trajectory.

• Line 167: 1000 iterations for an MCS is quite low. Are the authors confident that outputs stabilize after only 1000 iterations?

• Line 178: Could use a reference to show that biological optimum rotation cycle is 75 years.

• Line 184-186: The fate of the wood product strongly influences overall mitigation potential. It is not clear the relative allocations of biomass to unused mill residue, commercial fuel, and short-lived wood products or how those various pools are accounted for in the model. Can the authors provide more detail about what model inputs were (they don’t appear to be in Table S1) and justify why they chose those values?

• Line 199: The ambitious scenario extended rotations scenario is implemented on 40% of private timber lands and 32% of state-owned forest lands. Why 40 and 32%? What is the justification for this percentage?

• Line 204: what does a constant 10-fold multiplier mean? A 10% increase on the prior year’s rate? 10% of the ambitious scenario per year so that 40%/32% of forest land is on an extended rotation by year 2030?

• Line 208: The agricultural practices section is particularly sparse on details. There appears to be no limited, moderate or ambitious scenario (??). Further, the authors only provide data on baseline conditions for cover crops, nutrient use etc? What is the change in practice/the additional action modeled here?

• Line 221: The avoided conversion pathways have the same problem as the agricultural section. The authors only describe the baseline scenario and the baseline scenario (e.g., current cover crop adoption) is not a climate solution. A climate solution must represent additional action beyond the baseline. What change is being modeled here?

• Line 241: Restoration pathways have the same problem as above, though the tidal wetland restoration provides some detail (e.g., line 266, “We assigned each of those counties a restoration rate equal to its lost tidal wetlands hectares times a constant 0.00215”). Though again why 0.00215? Similarly line 270, “we obtained and inserted restoration areas described in existing literature” needs references.

Validity of the findings

I cannot evaluate the validity of the findings without understanding the methods. In particular I am concerned that the authors do not model additional action for many pathways but rather baseline scenarios.

Also an uncertainty analysis of the results would be helpful and seems possible given the MCS approach. There are additional tables in the supplementary material that appear to provide median and confidence intervals, but these are not referenced in the manuscript but only in a note to Peer J (Table S3-S7). Can the authors provide meta-data for these tables and refer to the uncertainty in the results within the manuscript itself?

Additional comments

Robertson et al. examine the mitigation potential of 11 different natural climate solutions in Washington State under two scenarios of adoption (medium and ambitious scenario). They then further break down potential opportunity by county to map where opportunities are highest for individual pathways, which is a particularly useful contribution. This is a timely study that represents an important bridge between science and policy. The county-level resolution is good for avoiding false certainty around opportunity at any specific location, but fine enough to inform decision making. The authors clearly put thought into developing their scenarios (e.g., excluding extended rotations in locations where wildfire risk is high, line 180). I believe that this paper has a strong potential to be an important contribution to the field, but currently there is not enough methodological detail to evaluate the results or the conclusions.

---

## Round 0.2 · Minor Revisions

Your manuscript can be considered to be accepted after addressing the minor comments from the reviewer. Please resubmit your revised manuscript as soon as possible. Thank you again for your submission to the PeerJ.

Reviewer 2 ·

Basic reporting

No comment

Experimental design

No comment

Validity of the findings

Since extended rotations show up as having the largest mitigation potential, I was glad to see that the authors included a lengthy discussion of leakage.

Additional comments

This is the second time that I have reviewed Leveraging the potential of nature to meet net zero greenhouse gas emissions in Washington State by Robertson et al. I remain convinced that this is a useful and informative study. The county-level disaggregation of results is particularly helpful for demonstrating how NCS opportunities vary across the state.

My prior concerns related mostly to insufficient methods documentation, which precluded evaluating the results and discussions sections. The methods (and introduction) are now much clearer in the revised draft so my final comments relate mostly to the results and discussions, which I did not review previously. Note that I read the track-change version of the manuscript so line numbers referenced below refer to that document rather than the cleaned-up PDF version. I would describe the below as “minor” edits.

Line 153: The 2020 to 2050 range is 31 years, if 2020 and 2050 are inclusive. 2021 to 2050 would be 30 years, in line with the reported results, and a 2021 start would also alleviate the issue of starting implementation in the past. Though I recognize that the data may be tied to a 2020 start, so a bit of clarification here would help. Is the modeling window 2020 to 2049?

Line 282: What is the motivation for clustering the agricultural activities (no-till, cover crops, and nutrient management)? This clustering biases the agricultural pathway towards showing up as a prominent opportunity in the results, simply because there are more activities baked into the calculations. Also, it assumes (I think?) that an individual farmer would be implementing all three. Is that assumption correct and are there any interactions between pathways that would impact the final estimates? For example, nitrogen fixing cover crops can reduce fertilizer needs. If cover crops reduce fertilizer needs, failing to account for this would lead to double counting with the nutrient management sub-pathway and inflate the estimate of mitigation potential. Similarly, aren’t cover crops typically plowed into the soil before planting and would that impact the no-till sub-pathway? This interaction could diminish the extent of opportunity for no-till. It would be helpful to include a sentence or two to justify this modeling assumption and clarify whether simultaneous adoption would inflate or alter the estimates.

Line 393 (typo): “8.9% of the reductions of annual emissions reductions needed in Washington” should be “8.9% of the annual emissions reductions needed in Washington”

Line 459: If rates of forest conversion to development have been declining continuously since 1991, is the modeled baseline a downward trend or a time averaged value? Table S2 suggests the latter, but its seems like the former would be more appropriate to avoid over-estimating additional mitigation potential. In other words, did the authors estimate the mitigation potential of avoiding actions that are unlikely to happen? A bit more justification in the methods would be useful.

Line 467: State should be capitalized? Inconsistent usage.

Line 472: This is not a “for example” given previous sentence. This sentence could start with “The greatest potential for emissions reduction occurs where restorable wetland…”

Line 685: “significant” is usually reserved for statistical significance, suggest “substantial” as an alternative

Line 690: As mentioned above by the authors, economic analyses are also needed to determine the cost-benefit ratio of any action

Table S1 dictionary (typo): conversion to urban AND rural development. Also, in other geographies the other dominant forest conversion vector is conversion to cropland. Does this not happen in Washington State?

Finally, there is one additional point that could be useful to include. Specifically, there is often a prevailing preference for restoration actions (e.g., UN Decade on Restoration, 1t.org, corporate pledges to only focus on negative emissions rather than avoided emissions). This paper is a useful demonstration of the relatively larger magnitude of opportunity from management rather than restoration actions. I see at least two places to make this point. First, Figure 3 is useful for demonstrating the counties with the largest opportunity for a given pathway, but misses an opportunity to demonstrate differences across pathways. Is it possible to adopt a single color scale across all panels to show these differences across all pathways (e.g., the darkest purple would refer to -0.65) or does that lose the very small range of opportunity for pathways like avoided conversion of grasslands? If yes, is there another way to use symbology to show both across county and across pathway variation? The other location to make this point is line 565-567 when the authors mention that forest restoration and afforestation alone are insufficient to nullify emissions.

---

## Round 0.3 · accepted · Accept

Your revised manuscript is accepted for publication by PeerJ. The article is well written. Thank you.